# Green Health Partnerships in Scotland; Pathways for Social Prescribing and Physical Activity Referral

**DOI:** 10.3390/ijerph17186832

**Published:** 2020-09-18

**Authors:** Sheona McHale, Alice Pearsons, Lis Neubeck, Coral L. Hanson

**Affiliations:** 1School of Health and Social Care, Edinburgh Napier University, Sighthill Campus, Edinburgh EH11 4DN, UK; A.Pearsons@napier.ac.uk (A.P.); L.Neubeck@napier.ac.uk (L.N.); C.Hanson@napier.ac.uk (C.L.H.); 2Sydney Nursing School, Charles Perkins Centre, Johns Hopkins Road, University of Sydney, Sydney, NSW 2006, Australia

**Keywords:** green health, social prescribing, public health, physical activity referral, qualitative evaluation

## Abstract

Increased exposure to green space has many health benefits. Scottish Green Health Partnerships (GHPs) have established green health referral pathways to enable community-based interventions to contribute to primary prevention and the maintenance of health for those with established disease. This qualitative study included focus groups and semi-structured telephone interviews with a range of professionals involved in strategic planning for and the development and provision of green health interventions (*n* = 55). We explored views about establishing GHPs. GHPs worked well, and green health was a good strategic fit with public health priorities. Interventions required embedding into core planning for health, local authority, social care and the third sector to ensure integration into non-medical prescribing models. There were concerns about sustainability and speed of change required for integration due to limited funding. Referral pathways were in the early development stages and intervention provision varied. Participants recognised challenges in addressing equity, developing green health messaging, volunteering capacity and providing evidence of success. Green health interventions have potential to integrate successfully with social prescribing and physical activity referral. Participants recommended GHPs engage political and health champions, embed green health in strategic planning, target mental health, develop simple, positively framed messaging, provide volunteer support and implement robust routine data collection to allow future examination of success.

## 1. Introduction

Globally, increasing numbers of people are living with non-communicable diseases, including heart disease, diabetes, cancer and mental health disorders [1]. Finding cost-effective and acceptable interventions to address primary prevention and maintain health for those with established disease is therefore a priority. The physical environment is a recognised determinant of health [2], and consequently, using the natural environment to promote good health is of increasing importance.

Increased exposure to green space is associated with reductions in all-cause mortality (31%), cardiovascular mortality (16%), type 2 diabetes (28%), and increased self-reported good health (12%) [3]. There is relatively strong evidence for the mental health and wellbeing benefits of being outdoors, including reductions in psychological stress, fatigue, anxiety and depression [4]. Those who report better access to green or recreational space also report narrower socioeconomic inequalities in mental well-being [4,5,6], meaning that encouraging people from deprived areas to access green space may contribute to health equity. However, improving access to green space has been shown to be a major concern for people living within cities across Europe. Challenges to accessing urban green space include availability being restricted to more affluent social groups, the attractiveness of green space not appealing to all, (for example, children), and accessibility limitations for specific groups such as the elderly or people with disabilities [7]. The benefits gained from being able to access the natural environment have potential to improve health by achieving recommended levels of physical activity (PA) in green space. Sufficient PA is associated with reductions in cardiovascular mortality and all-cause mortality [8], diabetes [9], breast cancer [10] and colon cancer [11], while also promoting social interactions and social equity [12,13].

Mechanisms are required to promote the use of the natural environment to improve health and increase PA undertaken in green space. Access to and use of the natural environment via referral by health and social care professionals to improve people’s health is defined as green health in this study [14]. Two established pathways exist that could be utilised to allow healthcare professionals (HCPs) to refer patients to green health interventions. Social prescribing schemes provide a pathway for referring patients from primary care services to non-clinical services in the community, often provided by third sector organisations [15]. Evidence for social prescribing is generally weak [16,17] and studies are required to better understand implementation. Physical activity referral schemes (PARS) are an internationally recognised pathway for referring patients from primary and secondary care to supervised PA opportunities [18,19]. Leisure providers predominantly deliver such schemes [20], mostly via indoor fitness classes or in gyms. There is a lack of studies examining whether PARS can be utilised in the natural environment. What is known is that if pathways for green health referral can be established, interventions encouraging access to, engagement with, or that use the environment as a setting to promote health result in positive impacts on quality of life, walking behaviours and mental health [21].

Little is known about the effective establishment of green health referral pathways. Scottish Green Health Partnerships (GHPs) are co-ordinated cross-sector partnerships including a range of organisations from health, social care, transport, education, sport and the environment sectors [14]. GHP’s aim to encourage use of the natural environment and establish pathways through which green health interventions can contribute to health improvement. Understanding the views of key individuals during the establishment of GHP’s can provide valuable insight into the implementation of green health into referral pathways [17]. In this study, we differentiated between two key groups of individuals: operational and strategic stakeholders. Operational stakeholders refer to those involved in the development and provision of green health interventions, improvements to green health and those who referred into established pathways (e.g., GHP project officers, education and health representatives, local authority environment department employees and leisure trust/third sector providers of referral schemes and interventions). Strategic stakeholders refer to those able to facilitate change at a policy, funding and strategy level (e.g., directors of public health, health and social care leads and community planning partnership leads). This study aimed to explore the views and experiences of a range of professionals involved in the establishment of GHPs, in order to gain insight of the influences on the establishment of pathways to promote green health.

## 2. Materials and Methods

The study employed a qualitative design, using focus groups for operational level participants and semi-structured telephone interviews for strategic level participants, to evaluate the experiences and views of key professionals involved in the establishment of GHPs across four Scottish health boards. The objectives were to explore the views of four GHP project officers, a number of GHP steering group members and f14 key senior strategic stakeholders of their experiences in establishing GHPs and referral pathways for green health. We report results using the Consolidated criteria for Reporting Qualitative Research [22].

### 2.1. Context

Scottish Natural Heritage is the lead organisation in the Our Natural Health Service programme [14], which aims to make better use of Scotland’s natural environment to improve health and wellbeing and tackle health inequalities. The programme hopes to encourage more people to enjoy and be active outdoors, to mainstream green health into health and social care policy and practice and to build capacity and participation within communities. It was not within the remit or budget of the GHPs to address the enablement of policies which would secure equity in green spaces availability and quality. Four GHPs were established during 2018/19 to demonstrate how to translate Scotland’s public health priorities [23] into practical action to encourage the use of green interventions to improve health. Each GHP area has a strategic lead, a steering group and a GHP project officer to coordinate developments. With three-year funding, GHPs are tasked with raising awareness of green health across key policy sectors and developing and strengthening links and referral pathways between health and social care and green health projects and providers. The lead partners in each area are the National Health Service (NHS) and local authority.

Edinburgh Napier University School of Health and Social Care research and integrity committee gave ethical approval for the study (REF: SHSC20003).

### 2.2. Sample

The qualitative study recruited a purposeful sample identified by Scottish Natural Heritage. The qualifying criteria included GHP project officers, GHP steering group members and key strategic stakeholders with knowledge and experience of the GHP’s. Scottish Natural Heritage sent GHP stakeholders information about the study prior to recruitment and requested permission to share contact details with researchers. Following this, the study team were provided with details of GHP project officers, steering group members and key strategic stakeholders. One researcher (SM) distributed invitations, participant information and consent forms to potential participants via email (*n* = 79). Recruitment took place over an eight-week period. We issued reminder emails at two weekly intervals. All participants gave voluntary written, signed consent.

### 2.3. Data Collection

Prior to commencement of the study, two researchers with qualitative experience (CLH, seven years, and SM, three years) met with Scottish Natural Heritage staff to conceptualise study design. After reviewing relevant literature, we developed interview guides (Appendix A) via a process of iterative review (one face-to-face meeting, followed by email and telephone review). Topics included support, facilitators and barriers for development of the GHPs, benefits, addressing health inequalities and future challenges.

GHP leads arranged the location and times for focus group interviews, and researchers contacted strategic participants via telephone and/or email to arrange convenient times for individual telephone interviews. Five focus groups (one for each GHP area steering group and one for GHP project officers) took place during December 2019, and semi-structured telephone interviews took place between November 2019 and January 2020. Three researchers (CLH, SM and AP) facilitated focus groups and two (SM and CLH) conducted telephone interviews. We conducted focus groups in private meeting rooms: three in NHS hospitals, one in a country park and one at the Edinburgh Napier University. We conducted 14 semi-structured telephone interviews from a private room at the university.

We consistently applied semi-structured interview guides across focus group and telephone interviews. For both focus groups and telephone interviews, we opened discussion with broad questions before encouraging and prompting participants to explore personal and shared perspectives. The focus groups/telephone interviews were recorded using an encrypted digital recorder. We made extensive field notes during and after individual/group discussions to supplement recordings, which focused on the quality of interactions and researcher bias. Due to geographical location and travel restrictions, we conducted one additional face-to-face semi-structured interview and one follow-up telephone interview with two GHP officers. An external agency transcribed audio files intelligent verbatim.

### 2.4. Data Analysis

We imported focus group and interview transcripts and field notes into NVivo12^®^ (QSR International, Melbourne, Australia) for analysis. We used numbering to replace participant names and GHP areas. After familiarisation with data and in accordance with methods of thematic analysis [24] using the framework approach [25], one researcher (CH) inductively created open codes (*n* = 37) for six telephone interviews. A second researcher (SM) independently analysed one transcript to check for consistency before open coding three more transcripts, including one focus group. At a team meeting, all researchers discussed developing themes, compared these with field notes and agreed an initial framework for analysis of other transcripts, which consisted of six themes (key partnerships, change, green health development strategies, the GHP officer, interventions and evaluation) and 13 subthemes. One researcher (SM) then deductively coded all other transcripts using the established framework. Prior to final analysis, two researchers (CH and SM) reviewed the framework and agreed two final overarching themes and eight subthemes.

## 3. Results

Participants (*n* = 55), 18 males and 36 females, included five GHP project officers (due to staff changes), 36 steering group members and 14 senior strategic participants (Table 1).

A range of professions, GHP project officers, directors of public health, social prescribing leads, general practitioners, Scottish Natural Heritage employees, local authority environmental officers, leisure providers and third community and voluntary sector representatives took part. Median focus group duration was 92 min (range 84–114 min) and median duration of telephone interviews was 47 min (range 31–78 min).

Participants described how GHPs had added value to the green health agenda and considered green health interventions had the potential to work via social prescribing and PA referral. We identified two overarching themes, strategic and operational factors, which required focus to integrate green health into non-medical prescribing models. Overall, green health was considered a good strategic fit but was not yet fully embedded into core planning for health, local authority, social care and the third sector. Participants were concerned about speed of change and sustainability given time-limited funding. At an operational level, GHP project officers were considered key in developing partnerships. Green health referral pathways were in the early development stages and intervention provision varied. Participants highlighted challenges in addressing health equity, developing green health messaging, volunteering capacity and in providing evidence of success (Figure 1).

### 3.1. Strategic Factors

Participants perceived that green health could contribute to public health priorities. GHPs had increased awareness in organisations such as health, education and the third sector, resulting in enthusiasm to develop links to interventions and embed green health in strategic pathways. However, participants expressed concerns about the cultural changes required for success given the limited three-year time frame.

#### 3.1.1. Strategic Fit

Key partners identified included the NHS (public health, primary and secondary care), local government (social care and environmental departments), leisure providers, the environmental sector (e.g., Forestry Commission Scotland), and the third, voluntary and community sector. Not all GHPs had the same partners; for example, education was an important partner in one area, while another highlighted the absence of social care from the steering group.

Participants perceived that green health interventions could contribute to all six Scottish public health priorities [23] by supporting prevention and health and wellbeing promotion. Participants reported that green health activities supported “*policy drivers for Realistic Medicine*” (*telephone interview participants 1 and 2*) and emphasised the potential role of green health in a “*non-pharmacological pathway*” (*telephone interview participant 10*). Previous green health initiatives developed over several decades meant that GHPs were “*pushing at an open door*” when developing strategic links. However, participants felt that initiatives, such as the Green Exercise Partnership [26], had made limited progress because “*It wasn’t funded. It wasn’t mandated. It was a partnership that had really come about at a local level through the kind of goodwill of partners*” (*telephone interview participant 6*). GHPs were considered more likely to achieve progress due to three-year funding and dedicated project officers.

Despite positive comments about progress and funding, participants suggested that inclusion of green health within strategic pathways was in its infancy and raised concerns about sustainability beyond the three-year funding period. They reported the need to embed green health activities into core planning for health, local authority, social care, and the third, voluntary and community sector. To encourage sustainability, participants suggested focusing on capacity building, using existing networks to target health inequalities and building in mechanisms to support community groups. To improve prominence, several participants discussed “targeting” one public health priority, suggesting GHP activities might increase in strategic importance if directed towards “*mental health issues … taking away the medical model from living with mental health issues*” (*telephone interview participant 10*). Additionally, participants suggested engaging HCPs and politicians as green health champions locally and nationally (*telephone interview participants 4 and 8; GHP Areas 1, 3 and project officer focus groups*).

#### 3.1.2. Cultural Change

Participants perceived that NHS culture change was important in developing green health initiatives. This included more proactive prevention, understanding the value of being outdoors, providing non-clinical interventions and accepting the role of the third sector in providing low-cost, informal activities. The requirement for cultural change applied to both HCPs and patients/service users.
Just getting across the idea that you might go into the doctor’s for a physical ailment or mental health issue, and that being referred to a gardening group is okay… There is a whole cultural thing that needs changing between patients, the wider community, the doctors, the occupational therapists, and all the referral partners (telephone interview participant 7).

Some positive changes were evident. One participant explained that she could now compliment previous lifestyle conversations with a green health prescription alongside other treatment (*telephone interview participant 9*). However, another indicated that she did not “*hear an awful lot about how clinicians see that as part of their role*” (*telephone interview participant 4*).

Cultural change involved all NHS staff having greater awareness of, and the ability to have positive conversations about, green health. For example, one GHP area upskilled GP receptionists to have exploratory conversations with patients to introduce ideas around social prescribing and signpost to link workers who could connect patients to community groups (*telephone interview participant 4 and 9*).

Participants highlighted that expecting NHS cultural change in three years was unrealistic. Expectations did not take account of reduced resources and staff asked to do more with less (*telephone interview participant 7*). However, participants reported that NHS early adopters were interested to test the promotion of green health, provide robust evidence of success (*GHP Area 4 steering group participant*), and explore ways to engage HCPs in green health promotion (*telephone interview participant 5*). Participants who had engaged with NHS staff discussed the time required to build trust, confidence and provide reassurance that third sector interventions would not suddenly disappear overnight.

### 3.2. Operational Factors

Key operational elements included the establishment of referral pathways, marketing of green health interventions, work force development and monitoring and evaluation of activity. GHPs had strengthened green health networks and given third sector, community and voluntary organisations a “*place at the table*” with health and local authority partners. Previously, such organisations had not necessarily perceived themselves as a health resource. Third sector partners reported benefits such as greater knowledge of community partners, improved access to information about local green health interventions and improved ability to signpost. Successes included positive engagement with HCPs, volunteers and partner organisations, while challenges included volunteer capacity, developing appropriate messaging to promote green health activities, improving activity inclusivity and capturing evidence of success.

#### 3.2.1. The Role of the GHP Project Officer

Participants considered GHP project officers a valuable resource and central to green health development. Although GHP areas had different employment models (one in local authority, two in NHS public health teams and one in the third sector), it was highlighted that the skills of the post holder were more important than who hosted the post, “*It’s about leadership and about being heard*” (*telephone interview participant 10*).

Sustaining the GHP officer role beyond the initial funding period was a subject of much discussion. Participants in one area considered this unlikely and that in a “no funded officer” scenario, individual green health projects would continue, but there was a danger that the HCPs would no longer engage. In an alternate view from another GHP area, one participant indicated that “*if in five years’ time a lot of the barriers have been broken down between NHS and green space services and third sector… and a lot of those connections have been made, we wouldn’t need a funded project officer*” (*telephone interview participant 6)*. This was qualified with a statement that they could not imagine this being the case.

#### 3.2.2. Establishing Referral Pathways

All areas had created new or strengthened existing pathways for green health referral. This work was twofold: ensuring that appropriate interventions existed for referrals and engaging health and social care professionals in the referral process. One important discussion was whether to create new pathways or use existing PARS/social prescribing pathways. Different approaches were evident.

One GHP area established a new “green prescription” scheme targeting people living in areas of high deprivation, which gave telephone advice about green health opportunities via an information hotline. There was an awareness of, but no integration with, social prescribing services. Another GHP area was in the process of setting up pilots, but a key criterion for selection was the presence of social prescribing link workers. Two GHP areas had integrated green health referrals into existing PARS pathways. Participants reported that although systems were in place, there had been few green health referrals:
Our referral form now has a category for walking/outdoor activity on it. I think it’ll take a bit of time for us to see if there is any return on that. (…) In the last quarter, we had 1978 [referrals], there were only 71 to walking (GHP Area 3 focus group).

Participants from one GHP area reported that using the existing PARS system for green health seemed like the most logical approach, but in reality, most referrals chose to attend sessions run by the PARS provider, who indicated that the scheme provided a new customer pool for their services. PARS staff had reservations about signposting to green health opportunities due to a lack of knowledge about sessions and concerns about whether deliverers of green health interventions had sufficient knowledge to deal with people considered higher risk. To mitigate these concerns, the PARS staff “*visited some of the green health interventions to improve understanding and increase confidence about signposting*” (*GHP Area 4 focus group*).

#### 3.2.3. Green Health Messaging

Participants discussed several aspects of messaging: communicating the role and purpose of the GHP to health and social care professionals and policy decision makers and raising public awareness of green health.

Steering groups acted as one mechanism to communicate with local policy decision makers. Many steering group members were from an operational background, but some were able to facilitate links with senior decision makers. GHPs had made a concerted effort to communicate with health and social care professionals, including personal visits to GP surgeries and organising events to establish and promote green health interventions:
One of our big successes was a green health event (…) we had over 170 health and social care professionals (…) healthcare support workers, social workers, nursing staff and consultants. (…) Green health providers offered taster activities, and they went round each of the different activities to see what we meant by green health (GHP Area 3 focus group participant).

Participants considered that promoting green health to the public was complex and expressed concerns about low health literacy levels. One participant suggested using the Scottish Government health literacy plan [27] when developing public messaging to ensure understanding of what green health opportunities were available, the benefits of engaging and how to access what existed (*telephone interview participant 5*). There were further concerns about the medicalisation of language; “health referral” and “green prescription” were deemed a potential barrier to the public and volunteer providers. Discussions centred on the concept of “health by stealth” and understanding potential participants’ motivation rather than “preaching” the benefits of being active outdoors. Additionally, participants reported the need to change perceptual barriers surrounding the use of green space:
There is endless research that tells us why people don’t use green spaces, because they feel unsafe, they don’t feel welcome. (GHP Area 3 focus group participant).

#### 3.2.4. Volunteering and Capacity Issues

Participants in some areas expressed concerns about the removal of public sector funding from community groups, while expecting such groups to accept health referrals and scale up interventions to meet demand. Delivery of green health interventions was mostly by third sector organisations, which tended to be small and reliant on volunteers. This created potential capacity issues, around volunteer numbers and the responsibility they were willing to accept. Barriers to volunteering included family/work commitments, transport, obtaining Protecting Vulnerable Groups checks and the need to commit to regular times. Community and third sector participants raised concerns that formalising referral pathways would require additional volunteer skills and increase the need for insurance, health and safety, and monitoring of outcomes.
They maybe don’t want to take official referrals (…). No, we’re actually quite happy just ticking along as we are (…). We’re not interested in doing, like, medical or health… official health based… we just, kind of, do what we want to do (GHP Area 2 focus group participant).

To counteract some of these issues, one GHP area had attracted additional funding to employ a green health volunteer development officer who co-ordinated and supported the network.

#### 3.2.5. Addressing Inequalities

Reducing health inequalities was core to the GHP model. Participants at a strategic level in one GHP area reported that their health equity strategy took a holistic view of health, taking into account alternatives to traditional healthcare and co-producing community programmes (*telephone interview participant 5*). Operationally, all GHPs reported targeting interventions in deprived areas. However, participants voiced concerns that locating an intervention in an area did not necessarily mean that it attracted local participants and that work should focus on target populations:
Otherwise, it’ll be all the young families with cars who say, oh love those community woods, the forestry’s done a brilliant job. Here, look at all the activities that the children adore, they come home with nice carved rabbits (…) And you’re thinking do you know what; they would have done it anyway (GHP Area 2 focus group participant).

All project areas mentioned challenges with green space assets and the need to improve accessibility and inclusivity for all. Accessibility issues included green space not being local to deprived areas and/or lacking public transport, and a lack of toilet facilities, cafes, self-medicating facilities, benches and adequate lighting. To address this, one GHP area reported mapping quality of green space alongside poor health indicators and seeking to make improvements where possible.

#### 3.2.6. Evidence and Evaluation

Each GHP area was completing a national evaluation, collecting quantitative data such as intervention numbers and qualitative data such as case studies. Participants perceived it difficult to capture evidence of behaviour change, other than uptake, attendance and adherence data (*GHP Area 3 focus group*). Since many community and voluntary groups did not keep registers, even these measures were difficult. One participant from education highlighted that the integration of data, such as the number of students taking part in green health activities and key performance indicators for student course adherence, could provide evidence of effect. Operational participants considered case studies a viable method for capturing success.

One GHP area discussed whether the key to securing future health funding was to provide randomised controlled trial evidence that investment in green health was a cost-effective way to improve health, despite the difficulties involved. Some strategic participants shared this view.
I am not really interested in boosting numbers of people contact with nature, it’s the right people and the impact it’s had… the financial benefit to the health practices (telephone interview participant 11).

At a pragmatic level, participants suggested that the role of the social prescribing link worker could be compared with usual care, using standardised quality of life measures (*GHP Areas 2 and 3 focus groups*) or questions from the Scottish Health Survey (*GHP Area 2 focus group*). Other suggestions included capturing reduction in medications, hospital re-admittance or re-offending rates for those who took part in green health interventions (*GHP Area 3 focus group*).

## 4. Discussion

Promoting use of the natural environment was considered a good strategic fit with Scottish public health priorities. Key partners were the NHS (public health teams and HCPs); local authorities (social care and environmental department); leisure providers; the environment sector and the voluntary, community and third sectors. GHPs and project officers in particular provided a powerful voice to raise the profile of green health, strengthened networks and gave third sector organisations a “place at the table” with health and local authority partners. Green health interventions have the potential to work as part of social prescribing and PARS pathways, although integration was in the early stages and provision varied. Participants identified the need to embed green health into core planning for partner organisations and recommended that focusing on mental health benefits could increase strategic importance. At an operational level, challenges highlighted were addressing equity of access, developing messaging, volunteering capacity and providing evidence of success.

To ensure longer-term sustainability and improve equality of access to green spaces, green health must be integrated into high-level strategic plans. Similar to evidence from across the globe, cross-departmental and governmental activity is necessary to realise the benefits of effectively using natural environments to improve health by encourage society to spend more time outdoors [28,29], but this is challenging. One approach that has proved successful with other health agendas is engaging senior politicians, HCPs and policy makers as champions [30,31]. In the case of green health, having higher-level representation could create momentum to allow referral pathways and interventions to evolve. This must be combined with addressing funding issues. In Scotland, integrating green health interventions with social prescribing has the potential to solve funding concerns. This is because the Scottish Government has committed that by 2021–2022 over half of NHS spending will be in community settings and recommended that social prescriptions be treated equally to medical prescriptions when issued by health and social care professionals [32]. This large shift over a short period is challenging.

We identified a need for cultural change in healthcare settings to convince HCPs that social prescribing improves health and wellbeing, given that they have an integral role in promoting green health to patients. Persuading patients that social interventions are an acceptable alternative to a medical prescription is a complex issue, but HCPs must reframe patient beliefs about suitability as this is known to contribute to successful uptake [33]. Furthermore, a cohesive and integrated healthcare approach to social prescribing is necessary to achieve community wide impact, yet this study and others have identified a need to improve communication and trust between medical prescribers and social prescribers and deliverers [15,34]. PARS demonstrate that patients are willing to accept “social prescriptions”, being a well-established pathway for HCPs to encourage patient PA [19].

Green health interventions could benefit from learning from PARS that “one size does not fit all” and that a menu-based approach, with consideration of participant motivations and needs, is more likely to result in success [35,36]. However, we discovered a lack of trust from PARS providers for green health interventions due to poor understanding and concerns regarding safety for higher risk referrals. There was a potential conflict of interest in PARS making onward referrals to green health interventions, as PARS success criteria tend to be uptake and retention [37], leading to the self-promotion of provider activities. For integration to work with PARS pathways, better understanding of, and alignment with, community-based PA options is required, and success must be defined by more than uptake and adherence.

Integrated community-based activities are uniquely placed to support the needs of people living in socio-economically deprived areas. Participants highlighted the impact of poor health literacy levels when targeting health beliefs surrounding green health. Overcoming health literacy limitations is an ongoing challenge to health promotion [38] and although appropriate health messaging can help address this, the best approach remains unclear. For example, messaging can increase knowledge and/or motivation, be fear-based (if you are not active, your health is threatened), stealth-based (have fun with your family in the park) or offer a reward (attend our allotment sessions and receive free fruit and vegetables). Participants in this study recommended a stealth-based approach and avoidance of medicalised terminology such as “prescription or referral”, which they considered could intimidate the public and volunteers. What is known is that public health messaging should be short, use simple language, be positively framed and stress the short-term benefits of participation [39].

Delivery of green health interventions is mostly by third sector, voluntary and community organisations. These tend to be small groups reliant on volunteers. In addition to delivering interventions, volunteers can offer extra support to those least able to access social situations, which is known to increase adherence [35,40,41]. Study participants expressed concerns about the changing responsibilities of green health volunteers with the implementation of formalised referral pathways and volunteer numbers required for scaling up delivery. Issues with volunteers included recruitment and retention, willingness to accept responsibility and accountability for participant health, and the boundaries of volunteering roles. To help address these issues, one GHP area employed a paid volunteer co-ordinator. Programme co-ordinators can be central to volunteers’ successful experience by setting safe boundaries, allowing volunteers to discuss client wellbeing, resolve disputes and arrange ongoing training [42]. Participants considered training particularly important to increase skills and confidence. To encourage optimal volunteer recruitment and retention, organisations should also match volunteers to the right role, provide clarity about how volunteers meet organisational objectives and quantify the value of volunteers [43].

Robustly defining what good looks like for nature-based interventions remains an ongoing challenge [44]. Traditionally, evidence of success requires control group comparison, long-term follow up and validated measuring tools [44,45,46]. However, the complexity of green health interventions means that exploring what works, for whom and in what circumstances [47,48,49] may better demonstrate success. Evaluation of routinely collected data may also provide evidence of success. For example, reduction in healthcare usage is the primary outcome measure most often suggested to assess the benefits of social prescribing [46]. Such measures require robust recording of referral and adherence in primary care systems. To enable examination of effect on health inequalities, referral and adherence data must link to a measure of socio-economic status. Participants in our study suggested focusing on mental health and so to examine success, interventions should consider including individual outcomes such as improvements in self-esteem, confidence, psychological wellbeing, health-related behaviours and day-to-day functioning that have previously been used to demonstrate success in social prescribing [50,51].

### Strengths and Limitations

This study provides naturalistic insight for green health intervention development, which is essential to inform social prescribing and public heath practice. We note the use of purposive sampling via GHP funders Scottish Natural Heritage, which potentially created a positive sampling bias. However, we used multiple strategies to triangulate findings, with trustworthiness enhanced by the involvement of multiple interviewers, peer debriefing and multiple perspectives collected from different stakeholder groups [52].

## 5. Conclusions

This study demonstrated success in the creation of partnerships to develop green health interventions that can integrate with social prescribing and PARS pathways. This is a good strategic fit with Scottish public health priorities. We recommend that GHPs engage political and health champions to encourage NHS cultural change; develop simple, positively framed green health messaging; engage with the third sector to provide volunteer support and implement robust routine data collection to allow for future examination of success.

## Figures and Tables

**Figure 1 ijerph-17-06832-f001:**
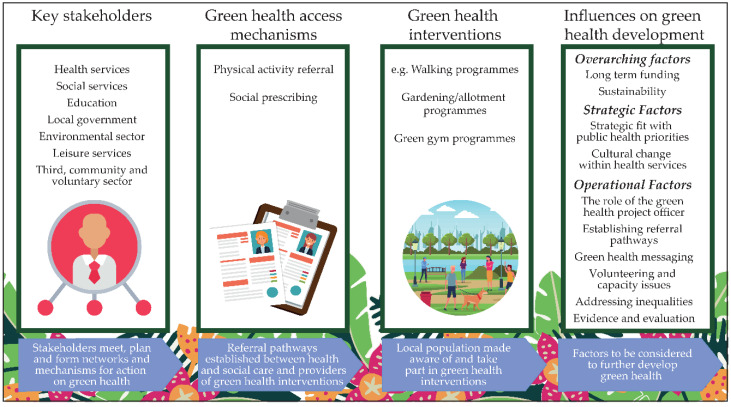
Strategic and operational factors in the development of green health interventions.

**Table 1 ijerph-17-06832-t001:** Focus and telephone interviews.

GHP Area	Number of Steering Group Members in Focus Group	Number of Strategic Participants Interviewed	GHP Project Officers Contributed Via
Area 1	8	4	Focus group
Area 2	6	1	Semi-structured interview
Area 3	9	5	Focus group and telephone interview
Area 4	13	4	Focus group

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
