# Peer review of "Green Health Partnerships in Scotland; Pathways for Social Prescribing and Physical Activity Referral"

_ijerph, 2020, doi:10.3390/ijerph17186832_

Round 1
Reviewer 1 Report
I think your team has an excellent idea and potentially important findings, but the lack of clarity in presenting the aims/objectives of your study and your methodology/methods obscures the importance and applicability of the idea.
For example, the title is too long. I would recommend shortening it, perhaps "Prescriptions for Physical Activity: A Study of Green Health Partnerships in Scotland" (or something to that effect).
As I read through the manuscript, I wondered if you were presenting findings from an evaluation rather than findings from a research process? If you are presenting evaluation findings, be transparent, it not - please clarify the aims, research questions/hypothesis and objectives of the study. Please be explicit so reader knows exactly what the aims/objectives are.
Also, define novel concepts you use. For example, what is "green health"? Where did that terminology emerge? What are "operational" and "strategic" stakeholders? etc. There are a few novel concepts throughout the manuscript which need clarification/definition/references.
Please clarify your methods. It's not enough to state you conducted a "qualitative" study. You had two data collection methods: focus groups and surveys. Describe your process of recruitment, the gender balance, age, etc (in a table?) and the duration of the focus groups. When you present "findings" - ensure that your participant quotations are indented. Be clear about the statistical analysis and how you programmed analytical software - and ensure that you discuss the limitations of the software.
Was your survey validated? If it was a survey for evaluation that was not verified and validated through a rigorous process of piloting, you may reconsider how you present it in the manuscript. As noted above, if you are presenting evaluation findings, the survey may provide a temperature reading. But please be transparent. You have a great concept/idea. Work on clarification of the research methods and define your novel terms throughout the Abstract/Intro.
Author Response
Thank you very much for your review of our manuscript. We set out our response to each point, and the resulting changes on the attached document.

Reviewer 2 Report
The study interestingly examines the Scottish example of green health referral pathways as a tool to improve the citizens' health and well-being. The study is well-planned and bases on surveys conducted with stakeholders. The information is then qualitatively assessed. Overall I find the results from the study which explores the effectiveness of employing physical activity referral schemes from the Green Health Partnerships.
The overall comments refer to in general quite a small sample size (n=55) and quite limited descriptions and analyses referring specifically to the Scottish system of health care and physical activity referral schemes. This is most visible in the abstract, which when read separately from the manuscript provides a significant number of details, while more attention should be given to reach an international audience. Who were the stakeholders? How were they affected by the study? This is unclear for the reader when only reading the abstract.
Lines-39-49. Exposure to green spaces has been recently examined in terms of environmental justice. I would expect the authors to show their overall knowledge in the matter and indicate that equal access to green spaces has been shown as a major concern in many cities, also inequalities for various vulnerable groups has become a key issue (although this is not the merit of this study), but from the point of view of this research, inequalities in access to green spaces by eg, children or senior are strongly linked to the study. Also availability of green spaces (the fact they exist) does not mean people actually use them, there is a good paper referring to this phenomenon I can recommend:
https://doi.org/10.1016/j.cities.2020.102862
I am not sure about the role of Figure 1 in the study, as much as I can appreciate the graphical idea, it is not clear how the various factors are interconnected, perhaps using a flow chart instead would be a better option
Are the authors able to present the results in any way quantitatively? Or at least draw general conclusions and group them to be presented in a tabular form. The results as they only consist of quotes from the surveys while they should be followed by systematic analysis. This should be presented as bullet points.
Please check the references, there are some missing links from the literature manager programme in the text, eg. Line 153.
I would also expect a broader analysis referring overall to green health
development but in terms of encouraging the society to spend time outdoors but also enabling policies which would secure the equity in terms of green spaces availability (also referring to the quality of those green spaces). The manuscript as it is is too narrow in my opinion.
Author Response
Thank you very much for your review of our manuscript. We set out our response to each point, and the resulting changes to the manuscript of the attached document.
